# Close Association between Awareness of Teeth-Alignment Disorder and Systemic Disorders in Late Adolescence

**DOI:** 10.3390/healthcare9040370

**Published:** 2021-03-26

**Authors:** Masanobu Abe, Akihisa Mitani, Atsushi Yao, Chun-Dong Zhang, Kazuto Hoshi, Shintaro Yanagimoto

**Affiliations:** 1Division for Health Service Promotion, The University of Tokyo, Tokyo 113-0033, Japan; mitania-int@h.u-tokyo.ac.jp (A.M.); yaoa-int@h.u-tokyo.ac.jp (A.Y.); yanagimoto@hc.u-tokyo.ac.jp (S.Y.); 2Department of Oral & Maxillofacial Surgery, The University of Tokyo Hospital, Tokyo 113-8655, Japan; hoshi-ora@h.u-tokyo.ac.jp; 3Division of Epigenomics, National Cancer Center Research Institute, Tokyo 104-0045, Japan; czhang@ncc.go.jp; 4Department of Gastrointestinal Surgery, Graduate School of Medicine, The University of Tokyo, Tokyo 113-0033, Japan

**Keywords:** teeth-alignment disorder, adolescence, systemic disease, malocclusion, periodontal disease, pollinosis

## Abstract

Background: Oral diseases are associated with various systemic disorders. Our previous research revealed new insights into the close relationship between occlusal disorder (functional disorder) and systemic disorders (allergic rhinitis, asthma, and arrhythmia) in late adolescence. Here, we investigated whether there was an association between the awareness of teeth-alignment disorder (morphological disorder) and common systemic disorders. Subjects and Methods: We retrospectively reviewed the data of the mandatory medical questionnaire that is required for the freshman medical checkup in Japan. We collected the data of all students who completed the questionnaire between April 2017 and April 2019. The data were analyzed using the *χ^2^* test, and a multivariate analysis was performed with a binomial logistic regression model. Results: The subjects were 8903 students aged 17–19 who had no awareness of occlusal disorder. The rate of awareness of teeth-alignment disorder was 20.43% (1819 of 8903 eligible subjects), and the aware students had significantly greater rates of gum bleeding (*p* < 0.001), pollinosis (*n* = 0.007), and atopic dermatitis (*n* = 0.042). The multivariate analysis revealed significant rates of gum bleeding (odds ratio (OR) 1.540, 95% confidence interval (CI): 1.386–1.711, *p* < 0.001), pollinosis (OR 1.197, 95% CI: 1.040–1.378, *p* = 0.012), and female gender (OR 1.141, 95% CI: 1.002–1.299, *p* = 0.046) among the students with awareness of teeth-alignment disorder. Conclusion: We identified close associations between the awareness of teeth-alignment disorder and both gum bleeding and pollinosis in a late-adolescent population. The systemic disorders that are targeted by teeth-alignment disorder were found to be different from those targeted by occlusal disorder.

## 1. Introduction

Oral diseases that are typified by dental caries and periodontal disease are associated with various systemic diseases and disorders including heart disease, diabetes, respiratory disease, rheumatism, metabolic syndrome, systemic infection, and malignant tumors [1,2,3,4,5,6,7,8,9,10]. However, the association between malocclusion and systemic diseases/disorders has not been focused on. Malocclusion is defined as a deviation from normal occlusion [11,12]. In this study, we classified the malocclusion into two types: a functional abnormality (occlusal disorder) and a morphological abnormality (teeth-alignment disorder) to assess the effect of malocclusion on general health.

Our previous research obtained new insight into the close relationship between an oral occlusal disorder (a functional abnormality) and common systemic diseases/disorders during adolescence including allergic rhinitis, asthma, and arrhythmia. These results not only reinforced the associations between occlusal disorder and both allergic rhinitis and asthma; they also demonstrated a new association between occlusal disorder and arrhythmia [13].

On the other hand, the impact of teeth-alignment disorder (a morphological abnormality) on systemic health has not been established. We conducted the present study to determine the association between the awareness of teeth-alignment disorder and common systemic disorders in late adolescents, even though the impact of teeth-alignment disorder on systemic diseases was considered to be smaller than that of occlusal disorder. Potential subjects who were aware of occlusal disorder were excluded from the analyses in order to remove the functional impact of occlusal disorder on the results.

## 2. Subjects and Methods

### 2.1. Study Design and Population

The completion of medical questionnaire is a legal requirement of freshman medical checkups (Students cannot have medical checkups without completion of the questionnaire). We retrospectively reviewed the data of this medical questionnaire completed between April 2017 and April 2019. The questionnaire is self-administered and consists of closed- and open-ended questions. The questionnaire was distributed to a total of 9376 students aged 17–19 during the specified period.

### 2.2. Questionnaire to the Students

The questionnaire was distributed to all freshmen aged 17–19 prior to the beginning of their medical checkups. The presence of the awareness of having teeth-alignment disorder was assessed by the question “Are you concerned about alignment of your teeth (appearance of teeth alignment)?” In the case of a “Yes” answer, the subject was categorized as having an awareness of teeth-alignment disorder (Appendix A). The presence of the awareness of occlusal disorder and the medical history were assessed by previously reported questions [13,14]. After evaluating the responses, the associations between the awareness of teeth-alignment disorder and systemic disorders were analyzed. Acute disorders and relatively rare diseases/disorders (i.e., those identified in <50 subjects) were excluded from the analysis [13].

### 2.3. Statistical Analyses

The data were analyzed using the *χ*^2^ test. We performed a multivariate analysis with the use of a binomial logistic regression model. A *p*-value < 0.05 (two-sided) was accepted as significant. We used the statistical software program SAS ver. 9.4 (SAS, Cary, NC, USA) and SPSS Statistics 25 (IBM, Armonk, NY, USA) for the analyses.

### 2.4. Ethical Approval

This study was approved by the research ethics committee of the University of Tokyo in 2018, approval no. 18-197 (revised as no. 19-324 in 2019).

## 3. Results

### 3.1. Frequency of Awareness of Teeth-Alignment Disorder

We retrospectively analyzed the data from 9098 students (aged < 20 years) of the 9376 who received the questionnaire [14]. Of these, we excluded 195 students who had awareness of occlusal disorder [13]. The final study population was 8903 students aged 17–19 years (mean 18.3 years) and included 7156 males and 1747 females. The rate of awareness of teeth-alignment disorder was 20.43% (1819 of the 8903 eligible subjects, 1438 males and 381 females). The rate of awareness of teeth-alignment disorder was higher among the females than the males, although the difference was not significant (*n* = 0.111) (Table 1).

### 3.2. Associations of Awareness of Teeth-Alignment Disorder with Systemic and Oral Disorders

The associations between the awareness of teeth-alignment disorder and 18 disorders in adolescence were analyzed. The disorders were as follows: 17 systemic disorders (pollinosis, food/drug allergy, inhaled antigen allergy, allergic rhinitis, otitis media/externa, sinusitis, pneumothorax/mediastinal emphysema, asthma/cough-variant asthma, atopic dermatitis, urticaria, scoliosis, spondylosis/spondylolisthesis/hernia, strabismus, myopia/hyperopia/astigmatism, arrhythmia, abnormal ECG other than arrhythmia, anemia) plus an oral disorder (gum bleeding) (Table 2). The awareness of teeth-alignment disorder was associated with a significantly greater incidence of a history of gum bleeding (*p* < 0.001), pollinosis (*n* = 0.007), and atopic dermatitis (*n* = 0.042).

The incidence of gum bleeding was significantly associated with the awareness of teeth-alignment disorder in both the male and female subjects (*p* < 0.001 in both genders). Pollinosis was significantly associated with the awareness of teeth-alignment disorder in only the female subjects (*n* = 0.007). Atopic dermatitis did not show a significant association in either gender.

The multivariate analysis using a binomial logistic regression model with teeth-alignment disorder as the objective variable (awareness of teeth-alignment disorder as an event) and the above-mentioned 18 disorders plus female gender as explanatory variables revealed significant rates of the following among the students with an awareness of teeth-alignment disorder: gum bleeding (odds ratio (OR) 1.540, 95% confidence interval (CI): 1.386–1.711, *p* < 0.001), pollinosis (OR 1.197, 95%CI: 1.040–1.378, *p* = 0.012), and female gender (OR 1.141, 95%CI: 1.002–1.299, *p* = 0.046) (Table 3).

When the subjects with awareness of occlusal disorder were not excluded from the analysis, the rate of awareness of teeth-alignment disorder was 21.05% (1915 out of 9098 subjects). These students had significantly greater rates of gum bleeding (*p* < 0.001), pollinosis (*p* = 0.007), and asthma/cough-variant asthma (*p* = 0.031). Multivariate analysis revealed significant rates of gum bleeding (OR 1.537, 95% CI: 1.386–1.703, *p* < 0.001), pollinosis (OR 1.195, 95% CI: 1.041–1.372, *p* = 0.011) and gender female (OR 1.139, 95% CI: 1.004–1.294, *p* = 0.044) among students with awareness of teeth-alignment disorder.

## 4. Discussion

The oral cavity and maxillofacial region are vulnerable to various diseases and disorders. In particular, dental caries and periodontal diseases are known to be associated with various systemic diseases [3,4,5,6,7,8]. In a previous study, we focused on an oral functional disorder, i.e., occlusal disorder, in late adolescence and observed close associations between occlusal disorder and systemic disorders including allergic rhinitis, asthma, and arrhythmia [13]. However, the relationships between teeth-alignment disorder, a morphological abnormality of the dental arch, with systemic disorders have not been focused on. Teeth-alignment disorder is sometimes accompanied by occlusal disorder. We therefore excluded the subjects with awareness of occlusal disorder from the present analyses in order to remove the functional impact on the results. Our findings revealed close associations between the awareness of teeth-alignment disorder with gum bleeding and pollinosis among late adolescents in Japan. Interestingly, the systemic disorders that are targeted by teeth-alignment disorder were found to be different from those targeted by occlusal disorder.

Several studies suggest that teeth-alignment disorder poses a risk of periodontal diseases, although this has not been established [15,16,17]. The present results clearly demonstrated a close association between the awareness of teeth-alignment disorder and bleeding gums, which implies that orthodontic treatment could improve periodontal health status and/or prevent the onset of periodontal diseases [18]. Certainly, adequate daily brushing is important to prevent periodontal diseases [19].

Allergic diseases/disorders are considered to be associated with malocclusion [13]. In this study, pollinosis was closely associated with teeth-alignment disorder in the present population. Although it is well known that allergic rhinitis (another common otorhinolaryngologic diseases in adolescence) causes occlusal disorder due to mouth breathing [13,20,21,22], the effects of pollinosis on dental disorders have not been clarified. As is a perennial/chronic disorder, allergic rhinitis could cause severe morphological disorder leading to the functional disorder by continuous mouth breathing [20,21,22]. In contrast, pollinosis (which is a seasonal allergic disorder) would not cause functional disorder but might induce a mild morphological disorder of the dental arch by temporary mouth breathing. Interestingly, we observed the significant association between pollinosis and teeth-alignment disorder only in the female subjects. Further research is required to examine this gender specificity.

Atopic dermatitis (which is an allergic disorder) was slightly but significantly associated with teeth-alignment disorder in our population. Perugia C et al. reported a higher prevalence of atopic dermatitis in pediatric dentistry patients compared to the general population; of the patients with atopic dermatitis, 64.4% had occlusal or alignment disorders [23]. Hannuksela et al. reported that atopic hyper-reactivity was a predisposing factor for posterior crossbites [24]. Recently, mouth breathing, which would cause malocclusion, has been reported as a risk factor for atopic dermatitis, even though the underlying mechanism has not been elucidated [25].

In the present multivariate analysis, female gender was an independent associating factor of teeth-alignment disorder. A difference in self-awareness and/or genetic or epigenetic backgrounds could be involved in this gender gap [26].

In summary, our retrospective analyses of the data of 8903 students aged 17–19 in Japan revealed close associations between the awareness of teeth-alignment disorder and both gum bleeding and pollinosis. Interestingly, associating disorders were different from those of occlusal disorder. These findings provide a foundation to obtain new evidence between malocclusion and systemic disorders. It can be said that orthodontic therapy may contribute to maintaining and improving the general health condition, even though these results were based on a self-report questionnaire completed by university students, and thus further research with clinical examinations is warranted.

## 5. Conclusions

We identified close associations between the awareness of teeth-alignment disorder and both gum bleeding and pollinosis in a late-adolescent population. The systemic disorders that are targeted by teeth-alignment disorder were found to be different from those targeted by occlusal disorder. Although further study is warranted to confirm the associations which were found here and explore the mechanisms underlying these associations, our results suggest that orthodontic therapy could contribute to maintaining and improving the general health condition.

## Figures and Tables

**Table 1 healthcare-09-00370-t001:** The rate of the presence/absence of awareness of teeth-alignment disorder in the study population.

Teeth-Alignment Disorder	Total (*n* = 8903)		Males (*n* = 7156)		Females (*n* = 1747)	
	*n*	%	*n*	%	*n*	%
Presence	1819	20.43	1438	20.10	381	21.81
Absence	7084	79.57	5718	79.90	1366	78.19

**Table 2 healthcare-09-00370-t002:** The association between awareness of teeth-alignment disorder and medical history.

Medical History	All			Male			Female		
	Awareness of Teeth-Alignment Disorder		*p*	Awareness of Teeth-Alignment Disorder		*p*	Awareness of Teeth-Alignment Disorder		*p*
	*n* (%)			*n* (%)			*n* (%)		
	Presence	Absence		Presence	Absence		Presence	Absence	
	1819 (100)	7084 (100)	–	1438 (100)	5718 (100)	–	381 (100)	1366 (100)	–
Pollinosis	322 (17.7)	1070 (15.1)	0.007 *	242 (16.8)	865 (15.1)	0.120	80 (21.0)	205 (15.0)	0.007 *
Food/drug allergy	56 (3.1)	210 (3.0)	0.859	44 (3.1)	173 (3.0)	0.946	12 (3.1)	37 (2.7)	0.775
Inhaled antigen allergy (except pollinosis)	43 (2.4)	133 (1.9)	0.217	34 (2.4)	105 (1.8)	0.234	9 (2.4)	28 (2.0)	0.862
Allergic rhinitis	287 (15.8)	1035 (14.6)	0.225	231 (16.1)	870 (15.2)	0.449	56 (14.7)	165 (12.1)	0.203
Otitis media/externa	43 (2.4)	133 (1.9)	0.217	36 (2.5)	110 (1.9)	0.199	7 (1.8)	23 (1.7)	1
Sinusitis	19 (1.0)	120 (1.7)	0.059	16 (1.1)	100 (1.7)	0.112	3 (0.8)	20 (1.5)	0.441
Pneumothorax/mediastinal emphysema	21 (1.2)	89 (1.3)	0.817	20 (1.4)	86 (1.5)	0.845	1 (0.3)	3 (0.2)	1
Asthma/cough variant asthma	192 (10.6)	647 (9.1)	0.071	157 (10.9)	560 (9.8)	0.222	35 (9.2)	87 (6.4)	0.073
Atopic dermatitis	148 (8.1)	477 (6.7)	0.042 *	120 (8.3)	396 (6.9)	0.071	28 (7.3)	81 (5.9)	0.372
Urticaria	21 (1.2)	107 (1.5)	0.304	15 (1.0)	90 (1.6)	0.169	6 (1.6)	17 (1.2)	0.806
Scoliosis	16 (0.9)	47 (0.7)	0.410	6 (0.4)	23 (0.4)	1	10 (2.6)	24 (1.8)	0.382
Spondylosis/spondylolisthesis/hernia	9 (0.5)	43 (0.6)	0.698	7 (0.5)	37 (0.6)	0.613	2 (0.5)	6 (0.4)	1
Strabismus	11 (0.6)	56 (0.8)	0.506	7 (0.5)	41 (0.7)	0.438	4 (1.0)	15 (1.1)	1
Myopia/hyperopia/astigmatism	10 (0.5)	72 (1.0)	0.085	7 (0.5)	57 (1.0)	0.093	3 (0.8)	15 (1.1)	0.807
Arrhythmia	24 (1.3)	75 (1.1)	0.412	19 (1.3)	61 (1.1)	0.496	5 (1.3)	14 (1.0)	0.842
Abnormal ECG other than arrhythmia	11 (0.6)	55 (0.8)	0.543	11 (0.8)	50 (0.9)	0.808	0 (0.0)	5 (0.4)	0.522
Anemia	17 (0.9)	43 (0.6)	0.173	9 (0.6)	22 (0.4)	0.308	8 (2.1)	21 (1.5)	0.594
Gum bleeding	809 (44.5)	2427 (34.5)	<0.001 *	661 (46.0)	2024 (35.4)	<0.001 *	148 (38.8)	385 (28.2)	<0.001 *

* *p* < 0.05.

**Table 3 healthcare-09-00370-t003:** Multivariate analysis for the association of awareness of teeth-alignment disorder with medical history.

Medical History	OR (95% CI)	*p*-Value
Gender (female)	1.141 (1.002–1.299)	0.046 *
Pollinosis	1.197 (1.040–1.378)	0.012 *
Food/drug allergy	0.954 (0.703–1.296)	0.765
Inhaled antigen allergy (except pollinosis)	1.119 (0.783–1.599)	0.539
Allergic rhinitis	1.085 (0.937–1.257)	0.273
Otitis media/externa	1.178 (0.828–1.675)	0.362
Sinusitis	0.587 (0.360–0.959)	0.034 *
Pneumothorax/mediastinal emphysema	0.941 (0.581–1.525)	0.805
Asthma/cough variant asthma	1.103 (0.924–1.317)	0.277
Atopic dermatitis	1.188 (0.974–1.450)	0.090
Urticaria	0.763 (0.474–1.229)	0.266
Scoliosis	1.288 (0.724–2.291)	0.389
Spondylosis/spondylolisthesis/hernia	0.799 (0.387–1.651)	0.545
Strabismus	0.748 (0.388–1.439)	0.384
Myopia/hyperopia/astigmatism	0.518 (0.265–1.012)	0.054
Arrhythmia	1.238 (0.776–1.977)	0.370
Abnormal ECG other than arrhythmia	0.785 (0.408–1.510)	0.468
Anemia	1.460 (0.825–2.585)	0.194
Gum bleeding	1.540 (1.386–1.711)	<0.001 *

* *p* < 0.05; OR = odds ratio; CI = confidence interval.

## Data Availability

Data is contained within the article or Appendix A.

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
