# Peer review of "Close Association between Awareness of Teeth-Alignment Disorder and Systemic Disorders in Late Adolescence"

_healthcare, 2021, doi:10.3390/healthcare9040370_

Round 1

Reviewer 1 Report

Dear Authors,

thanks for your manuscript. I found it very interesting and providing new insights into the sentinel role of orthodontists.

Here some concerns:

Background section

You define malocclusion as "an oral functional abnormality". There is a good amount of literature confirming that only certain malocclusions (openbite, crossbite, increased overjet) could be related to functional problems. I would change your definition as "a deviation from normal occlusion": a malocclusion could be defined as a morphological abnormality rather than a functional abnormality.

similarly : "teeth malalignment (which is an oral structural disorder)"

page 2, l6: there is a typo error "results2. Materials and Methods"

An hypothesis or a null hypothesis should be clearly stated closing the introduction section

Materials and Methods section

It could be of interest to know the population sample: they are students of which course? Are they students of the School of Medicine? or of Dental School? or were they laypeople?

Discussion

Again as mentioned above, sentences like "focused on an oral functional disorder, i.e., malocclusion," are not acceptable in 2021

"Teeth malalignment is sometimes accompanied by malocclusion": I think there is some confusion on the definition of malocclusion. Misalignment of teeth is a malocclusion, remember that you can have class I malocclusion with normal OJ and OB because of teeth misalignment.

Closing the discussions section with more emphasis on the clinical implications of your findings will add interest to your research

Author Response

We appreciate this reviewer's precise and constructive comments.

Comments:

  1. Background section. You define malocclusion as "an oral functional abnormality". There is a good amount of literature confirming that only certain malocclusions (openbite, crossbite, increased overjet) could be related to functional problems. I would change your definition as "a deviation from normal occlusion": a malocclusion could be defined as a morphological abnormality rather than a functional abnormality. similarly : "teeth malalignment (which is an oral structural disorder)"

This expert opinion is really heipful for us. Owing to the comment, we quit using the words “malocclusion” and “teeth malalignment”. Instead of them, we determined to use  “occlusal disorder” and “ teeth-aligment disorder”, respectively.

  1. Background section. page 2, l6: there is a typo error "results2. Materials and Methods"

We thank for this comment. We corrected the typing error.

  1. An hypothesis or a null hypothesis should be clearly stated closing the introduction section

The following description was added to Background section. "We conducted the present study to determine the association between the awareness of teeth-alignment disorder and common systemic disorders in late adolescents, even though the impact of teeth-alignment disorder on systemic diseases was considered to be weaker than that of occlusal disorder."

  1. Materials and Methods section. It could be of interest to know the population sample: they are students of which course? Are they students of the School of Medicine? or of Dental School? or were they laypeople?

We thank this reviewer's comment. They are laypeople. The following description was added to Subjects and Methods Section.  The questionnaire was distributed to “all freshmen” prior to the beginning of their medical checkups.

  1. Again as mentioned above, sentences like "focused on an oral functional disorder, i.e., malocclusion," are not acceptable in 2021. "Teeth malalignment is sometimes accompanied by malocclusion": I think there is some confusion on the definition of malocclusion. Misalignment of teeth is a malocclusion, remember that you can have class I malocclusion with normal OJ and OB because of teeth misalignment.

We thank this reviewer's critical comment. Owing to this comment, we modified Discussion section. we used  “occlusal disorder” and “ teeth-aligment disorder” instead of “malocclusion” and “teeth malalignment”, respectively.

  1. Closing the discussions section with more emphasis on the clinical implications of your findings will add interest to your research.

Thank you so much for this constructive suggestion. Owing to this suggestion, we emphasized the importance of orthodontic therapy for maintaining and improving general health condition in the closing the Discussion section as follows. “It can be said that orthodontic therapy may contribute to maintaining and improving the general health condition.”

Reviewer 2 Report

The manuscript is correctly written and shows coherence between the objectives and results. However, in my opinion, it shows deficiencies that make its publication questionable.

The authors relate the information obtained through a survey with organic
responses such as the immune response in the presence of allergens
(pollinosis), as well as with musculoskeletal changes.
Although the statistical analysis of the data is unquestionable,
the biological mechanisms of the possible association between the variables
is not shown.

The authors do not propose the biological and physiological mechanisms of how tooth misalignment increases the risk of pollinosis, or through what possible or putative mechanisms does pollinosis cause tooth misalignment?
Through what pathophysiological mechanisms do atopic dermatitis produce misalignment dental? 

For example, in the third paragraph of page 6 (Discussion) the assumption is not supported by citations, the authors must support the suggestions with scientific information, otherwise, there is a risk of being speculative.

Author Response

We appreciate this reviewer's critical comments.

Comments:

The biological mechanisms of the possible association between the variables
is not shown. The authors do not propose the biological and physiological mechanisms of how tooth misalignment increases the risk of pollinosis, or through what possible or putative mechanisms does pollinosis cause tooth misalignment?
Through what pathophysiological mechanisms do atopic dermatitis produce misalignment dental? 

We thank this reviewer's critical comment. Biological mechanisms of the relationship of tooth misalignment with pollinosis and atopic dermatitis were discussed as much as possible in the Discussion section.

Reviewer 3 Report

Authors should review the manuscript before considering publication:

1) The introduction is very short and with few bibliographical references.

2) There is no adequate description of the background of the study in the introduction.

3) There is an error in the last line of the introductory section.

4) Why was the questionnaire not included in the manuscript?

5) Why was an age range not included in the study sample?

6) How was it checked that the patients completed the questionnaire correctly?

7) In the discussion it is necessary to describe the main limitations of the study.

8) A conclusion section should be added to the manuscript.

9) It is necessary to revise the bibliographical references because they do not comply with the journal's regulations.

Author Response

We appreciate this reviewer's precise comments.

Comments:

  1. The introduction is very short and with few bibliographical references.

The volume  of  introduction and number of bibliographical references were increased.                                                                                                                                                                                                                                        

  1. There is no adequate description of the background of the study in the introduction.

Owing to this comment, Background section was modified.

  1. There is an error in the last line of the introductory section.

We thank for this comment. We corrected the typing error.

  1. Why was the questionnaire not included in the manuscript?

Questionnaire was prepared (Supplementary file 1).

  1. Why was an age range not included in the study sample?

The age range of study samples was between 17–19 years (mean 18.3 years).

  1. How was it checked that the patients completed the questionnaire correctly?

The completion of medical questionnaire is a legal requirement of freshman medical checkups in The University of Tokyo. Students can not get medical checkups without completion of the questionnarire. This was described in Subjects and Methods section.

  1. In the discussion it is necessary to describe the main limitations of the study.

The main limitations were described as follows in the Discussion section. “However, these results were based on a self-report questionnaire completed by university students, and thus further research with clinical examinations is warranted.”

  1. A conclusion section should be added to the manuscript.

Thank you so much for this comment. Conclusion section was added in the manuscript.

  1. It is necessary to revise the bibliographical references because they do not comply with the journal's regulations.

Owing to this comment, the bibliographical references were revised.

Round 2

Reviewer 2 Report

The authors responded to the reviewer's suggestions

Reviewer 3 Report

The authors have improved the manuscript.

This manuscript is a resubmission of an earlier submission. The following is a list of the peer review reports and author responses from that submission.